# Repurposing Niclosamide as a Novel Anti-SARS-CoV-2 Drug by Restricting Entry Protein CD147

**DOI:** 10.3390/biomedicines11072019

**Published:** 2023-07-18

**Authors:** Zhe Yang, Qi Zhang, Xiaoqing Wu, Siyuan Hao, Xinbao Hao, Elizabeth Jones, Yuxia Zhang, Jianming Qiu, Liang Xu

**Affiliations:** 1Department of Molecular Biosciences, The University of Kansas, Lawrence, KS 66045, USA; 2Higuchi Biosciences Center, The University of Kansas, Lawrence, KS 66045, USA; 3The University of Kansas Cancer Center, The University of Kansas Medical Center, Kansas City, KS 66160, USA; 4Department of Microbiology, Molecular Genetics and Immunology, The University of Kansas Medical Center, Kansas City, KS 66160, USA; 5Department of Pharmacology, Toxicology & Therapeutics, The University of Kansas Medical Center, Kansas City, KS 66160, USA; ejones11@kumc.edu (E.J.);; 6Department of Radiation Oncology, The University of Kansas Medical Center, Kansas City, KS 66160, USA

**Keywords:** CD147, RNA-binding protein, HuR, niclosamide, SARS-CoV-2, Western blot

## Abstract

The outbreak of severe acute respiratory syndrome coronavirus 2 (SARS-CoV-2) has led to the global coronavirus disease 2019 (COVID-19) pandemic, and the search for effective treatments has been limited. Furthermore, the rapid mutations of SARS-CoV-2 have posed challenges to existing vaccines and neutralizing antibodies, as they struggle to keep up with the increased viral transmissibility and immune evasion. However, there is hope in targeting the CD147-spike protein, which serves as an alternative point for the entry of SARS-CoV-2 into host cells. This protein has emerged as a promising therapeutic target for the development of drugs against COVID-19. Here, we demonstrate that the RNA-binding protein Human-antigen R (HuR) plays a crucial role in the post-transcriptional regulation of CD147 by directly binding to its 3′-untranslated region (UTR). We observed a decrease in CD147 levels across multiple cell lines upon HuR depletion. Furthermore, we identified that niclosamide can reduce CD147 by lowering the cytoplasmic translocation of HuR and reducing CD147 glycosylation. Moreover, our investigation revealed that SARS-CoV-2 infection induces an upregulation of CD147 in ACE2-expressing A549 cells, which can be effectively neutralized by niclosamide in a dose-dependent manner. Overall, our study unveils a novel regulatory mechanism of regulating CD147 through HuR and suggests niclosamide as a promising therapeutic option against COVID-19.

## 1. Introduction

Since its emergence in late 2019, the world has been engaged in an ongoing battle against COVID-19 caused by SARS-CoV-2. The global impact of the pandemic has been immense, with over 767 million reported infections and approximately 6.9 million recorded deaths as of June 2023 [1]. While the development and widespread administration of vaccines have provided a crucial defense against the virus, the rapid mutation rate of SARS-CoV-2 continues to present new challenges [2,3]. Despite the administration of over 13 billion vaccine doses worldwide, the constant threat of new variants looms [2,3].

Currently, only three drugs have gained approval from the U.S. Food and Drug Administration (FDA) for the treatment of SARS-CoV-2: Paxlovid, Remdesivir, and Molnupiravir. Paxlovid, consisting of Nirmatrelvir, an active 3CL protease inhibitor [4], and ritonavir, an HIV-1 protease inhibitor and CYP3A inhibitor [5], has been proven effective. Remdesivir and Molnupiravir, ribonucleotide analogue inhibitors of viral RNA polymerase, have also shown promise [6,7]. However, the currently available drugs have notable limitations, particularly for patients with renal disease [8,9,10]. Furthermore, the potential long-term effects of COVID-19, such as multisystem inflammatory syndrome, autoimmune conditions, pulmonary fibrosis, and myocarditis [11], are causes for significant concern as they may impact the quality of life for recovered individuals. Therefore, there is an urgent need for extensive research to identify new therapeutic targets and develop effective drugs to combat COVID-19 and address its associated complications.

CD147, also known as Basigin (*BSG*) or extracellular matrix metalloproteinase inducer, has emerged as a potential alternative entry point for SARS-CoV-2, in addition to Angiotensin-converting enzyme 2 (ACE2) [12,13]. The CD147-dependent entry mechanism is believed to involve Arf6 [14]. As a membrane receptor, CD147 plays various roles in tumor metastasis [15], progression [16,17], and viral infection [18]. Studies using human CD147 knock-in NSG mice have demonstrated increased susceptibility to SARS-CoV-2 compared to WT-NSG mice, further supporting the involvement of CD147 as a viral entry receptor [19]. Similarly, an iPSC-derived kidney podocyte model has identified CD147 as a key mediator of spike protein binding activity [20]. Analysis of differentially expressed proteins has revealed upregulation of CD147 in samples from COVID-19 patients [21]. Furthermore, a recent study analyzing patient platelet indices and COVID-19 transcriptomic signatures has shown that megakaryocytes actively internalize SARS-CoV-2 through CD147, and patient platelets exhibit a unique proinflammatory transcriptome and increased reactivity [22]. Another study has indicated CD147-dependent platelet activation upon interaction with SARS-CoV-2, underscoring the functional role of CD147 in the infection process [23].

The signaling mediated by CD147 has also been implicated in the disruption of cardiac pericytes by the SARS-CoV-2 spike protein [24]. Additionally, CD147 has been associated with SARS-CoV-2-induced pulmonary fibrosis, a significant post-COVID condition [25]. Antibody-mediated blocking of CD147 has shown promise in reducing viral gene expression in megakaryocytes [22]. Notably, a clinical trial targeting CD147 using the humanized anti-CD147 IgG2 monoclonal antibody meplazumab has demonstrated accelerated recovery in COVID-19 patients [26]. Furthermore, this antibody has exhibited effective inhibition of viral infection and the cytokine storm induced by SARS-CoV-2 and its variants [27]. These findings, coupled with the encouraging results from clinical trials, highlight the potential of CD147 as a therapeutic target for treating COVID-19.

Niclosamide, an FDA-approved anti-helminthic drug, has demonstrated remarkable effectiveness in inhibiting not only SARS-CoV [28] and MERS-CoV [29], but also SARS-CoV-2 [30] and its variants including Alpha, Beta, Delta, and Omicron [31,32]. The promising results have prompted multiple clinical trials investigating the safety and efficacy of niclosamide against SARS-CoV-2 (see Appendix A). Niclosamide exhibits its versatility by exerting suppressive effects on various targets. It inhibits the transcription factor STAT-3 [33], the calcium-binding protein S100A4 [34], the calcium-activated chloride channel protein TMEM16A [35,36], and several cancer-related signaling pathways. Additionally, niclosamide acts as a mitochondria uncoupler [37]. Recent findings have revealed that niclosamide inhibits TMEM16 proteins, thereby blocking virus spike-induced syncytia formation [35]. It also effectively suppresses inflammasome activation and restricts SARS-CoV-2 replication [38]. Moreover, niclosamide has emerged as a potential therapeutic option for alleviating pulmonary fibrosis by inhibiting the TGF-beta signaling pathway and its effectors, such as alpha smooth muscle actin and fibronectin [39,40,41,42,43,44,45].

In this study, our objective was to explore methods for modulating CD147, and we successfully identified niclosamide as an effective agent for reducing CD147 protein levels. Our findings indicate that the RNA-binding protein HuR (Human antigen R) binds to the 3′-UTR of *BSG* mRNA, and the depletion of HuR results in a decrease in CD147 protein expression. Niclosamide inhibits the translocation of HuR from the nucleus to the cytoplasm, thereby potentially suppressing the expression of *BSG* through various mechanisms. Notably, niclosamide effectively counteracted the SARS-CoV-2-induced upregulation of CD147. Overall, our study reveals the multifunctional inhibitory effects of niclosamide on CD147, providing evidence for repurposing niclosamide as a functional inhibitor of CD147 and as a potential therapeutic for COVID-19.

## 2. Materials and Methods

### 2.1. Cell Culture

Human breast cancer cell line MDA-MB-231, colon carcinoma cell line RKO, lung cancer cell line H460, human embryonal kidney (HEK) 293-FT cells, human fibroblast cell line WI-38, human lung bronchus epithelial cell line NL20, and mouse lung cancer cell line Lewis lung-2 (LL/2) were purchased from American Type Culture Collection (ATCC, Manassas, VA, USA). Human cervical cancer cell line Hela was kindly provided by Dr. Dan Dixon’s lab. NL20 cells were cultured in ATCC-formulated F-12K medium supplemented with 1.5 g/L sodium bicarbonate, 2.7 g/L glucose, 2.0 mM L-glutamine, 0.1 mM nonessential amino acids, 0.005 mg/mL insulin, 10 ng/mL epidermal growth factor, 0.001 mg/mL transferrin, 500 ng/mL hydrocortisone, and 4% fetal bovine serum (FBS, Sigma-Aldrich, Darmstadt, Germany, Cat# F4135). H460 cells were maintained with ATCC-formulated RPMI medium supplemented with 10% FBS, 1% penicillin–streptomycin (Corning, New York, NY, USA, Cat# 30-002-CI), and 1% L-glutamine (Corning, Cat# 25-005-CI). Other cells were cultured with DMEM medium with the same supplements. All cells were cultured in the incubator at 37 °C with 5% CO_2_. All cell lines were either recently obtained or monitored by short tandem repeat DNA profiling. A549-ACE2 cells: A549 cells stably expressing ACE2 (A549-ACE2) were generated by transducing lentivirus-encoded human ACE2 gene and were selected by 10 µg/mL blasticidin for 3 weeks [46]. A549-ACE2 cells were cultured in DMEM supplemented with FBS (10%), streptomycin (100 µg/mL), and penicillin (100 units/mL).

### 2.2. Chemicals and Reagents

Niclosamide was purchased from Calbiochem (Cat# 481909-1GM). For cell treatment, niclosamide powder was prepared in dimethyl sulfoxide (DMSO, Sigma-Aldrich, Cat# D8418) at a concentration of 10 mmol/L and further diluted in cell culture medium for indicated working concentrations for the treatment of cells. For the animal study, niclosamide powder was dissolved in PBS with 10% Tween-80 (Sigma-Aldrich, Cat# P4780) and 5% ethanol at a concentration of 2 mg/mL for intraperitoneal injection.

### 2.3. Immunofluorescence Staining and Microscopy Imaging

WI-38 and NL20 cells were cultured in a chamber slide (Lab-Tek II, Rochester, NY, USA, Cat# 154526) and treated with 1µM niclosamide or DMSO for 48 h. Then, cells were fixed with 100% methanol (chilled at −20 °C) at room temperature for 5 min. Cells were then washed with ice-cold PBS and permeabilized with 0.1% Triton X-100 (Sigma-Aldrich, Cat# T8787) for 10 min at room temperature. Cells were incubated with 1% bovine serum albumin (BSA, Fisher Scientific, Waltham, MA, USA, Cat# BP1605-100) and 22.52 mg/mL glycine (Sigma-Aldrich, Cat# G7126) in PBST (PBS with 0.1% Tween 20) for 30 min to block unspecific binding of the antibodies and then incubated with anti-HuR antibody (Santa Cruz, CA, USA, 3A2, Cat# sc-5621) at a 1∶200 dilution in 1% BSA overnight at 4 °C. Cells were then washed and incubated with anti-mouse IgG conjugated with FITC at a 1∶32 dilution in 1% BSA for one hour at room temperature. One μg/mL DAPI was then used for nuclear staining and was incubated for 1 min. Slides were mounted with SlowFade Gold antifade reagent (Invitrogen, Carlsbad, CA, USA, Cat# S36938) and DAPI containing mounting medium for fluorescence (Vector Laboratories, Newark, CA, USA, Cat# H-1200). Cells were imaged using an Olympus IX71 microscope using DP Controller and DP Manager software (3.3.1.222). Images were merged using ImageJ (1.53e).

### 2.4. Western Blotting

To extract total protein, cells were lysed in ice-cold 1X RIPA lysis buffer PMSF protease inhibitor, EDTA-free protease inhibitor cocktail (Roche Diagnostics GmbH, Basel, Switzerland, Cat# 11836170001), and phosphatase inhibitor cocktail (ThermoFisher Scientific, Waltham, MA, USA, Cat# 78426) on ice followed by centrifugation at 12,000 rpm and 4 °C for 20 min. NE-PER Nuclear and cytoplasmic extraction reagents kit (ThermoFisher Scientific, Cat# 78835) was used for nuclear and cytoplasmic protein extraction following the manufacturer’s instructions. Protein concentrations were measured by Bradford protein assay using protein assay dye (Bio-Rad, Hercules, CA, USA, Cat# 5000006). Lysate was heated for 5 min at 95 °C in SDS sample buffer, separated by SDS-PAGE, and transferred to a PVDF membrane. Membranes were blocked in 5% non-fat milk in TBST with 0.1% Tween and then probed with the indicated antibodies. The reactive bands were visualized using an Odyssey FC imaging system from LI-COR Biosciences. Antibodies: Mouse anti-EMMPRIN antibody (8D6, Cat# sc-21746), mouse anti-HuR antibody (3A2, Cat# sc-5621, 1:500 dilution), and mouse anti-GAPDH antibody (10B8, Cat# sc-51905, 1:1000 dilution) were purchased from Santa Cruz Biotechnology, USA. Anti-α-Tubulin (Cat# T5168, 1:2000 dilution) was obtained from Sigma-Aldrich. IRDye Goat anti-mouse IgG secondary antibodies (Cat# 926-68070 and Cat# 926-32210) and IRDye Goat anti-rabbit IgG secondary antibodies (Cat# 926-68071 and Cat# 926-32211) were purchased from LI-COR Biosciences and used at 1:10,000 dilution. Western ECL substance kit for HRP conjugate was purchased from Bio-Rad (Cat# 170-5061).

### 2.5. Ribonucleoprotein Immunoprecipitation and RNA Immunoprecipitation

For ribonucleoprotein immunoprecipitation (RNP-IP) assay, NL20, H460, and WI38 cells were cultured for 48 h and then lysed on ice using the Immunoprecipitation Kit (DynabeadsTM Protein G, Invitrogen, Cat# 10004D). HuR antibody or mouse IgG (BD Biosciences) was added to the cell lysate. After 1h incubation on ice, Protein G agarose was added to pull down HuR protein. Then, the RNAs bound to HuR were isolated using Trizol reagent, and the mRNA level was tested using RT-qPCR. For the RNA immunoprecipitation (RNA-IP) assay, cultured NL20 and WI-38 cells were lysed on ice using the Immunoprecipitation Kit (Dynabeads Streptavidin Trial Kit, Invitrogen, Cat# 65801D). The cell lysate was incubated with biotinylated *BSG* oligo (5′-CUUUUAUGUUUAAUU-3′, 1 µM, purchased from Horizon Discovery) or random RNA oligo (1 µM, purchased from Horizon Discovery) for 1 h with or without unbiotinylated *BSG* oligo (10 µM, purchased from Horizon Discovery). Biotin-labeled oligo and its bound HuR protein were immunoprecipitated using the DynabeadsTM Streptavidin Trial Kit (Invitrogen, Cat# 65801D). Western blotting was then performed to detect HuR protein.

### 2.6. Virus and Virus Infection

SARS-CoV-2 (NR-52281), isolate USA-WA1/2020, was obtained from BEI Resources, NIAID, NIH. The viruses were propagated in TMPRSSETMPRSS2-expressing Vero cells (Vero-TMPRSS2), titrated by plaque assay in Vero cells, aliquoted in D-PBS (pH 7.4), and stored at −80 °C [47,48]. The virus titer was 1.0  ×  10^7^ plaque-forming units (PFU)/mL. A549-ACE2 cells cultured in 24 well plates were treated with niclosamide at concentrations of 0, 0.25, and 0.5 μM. After 2 h, the cells were infected with SARS-CoV-2 at a multiplicity of infection (MOI) of 2. At 3 days post-infection, the cells were collected by centrifugation, followed by a wash with D-PBS. The cell pellets were resuspended in 1× Laemmli SDS sample buffer and boiled at 95 °C for 15 min. A biosafety protocol to work on SARS-CoV-2 in the biosafety level (BSL3) lab was approved by the Institutional Biosafety Committee of the University of Kansas Medical Center.

### 2.7. Animal Study

Female C57BL/6 mice aged 4–6 weeks purchased from Charles River Laboratories were used for efficacy studies. First, 2 × 105 LL/2 cells in 0.2 mL DMEM were inoculated to #3 mammary fat pad. Mice with best-matched tumors were randomized into two groups and subsequently treated with either PBS or 20 mg/kg niclosamide via intraperitoneal injection. The administration of niclosamide or PBS continued daily for 5 consecutive days. On day 6, tumor samples were collected and immediately fixed in 4% paraformaldehyde for IHC staining. Animal care and experiments were performed in accordance with the protocols approved by the Institutional Animal Use and Care Committee at the University of Kansas.

## 3. Results

### 3.1. HuR Binds BSG mRNA and Regulates CD147 Translation

In a previous study, blood samples were collected from COVID-19 patients and subjected to RNA sequencing, revealing a correlation between CD147 and the progression of COVID-19. The samples were divided into four groups: healthy, early-stage (COVID-19 positive for 0–10 days), middle-stage (COVID-19 positive for 11–20 days), and late-stage (COVID-19 positive for >20 days). Analysis of the RNA-seq data demonstrated a significant association between the expression of *BSG* (CD147 gene) and the advancement of COVID-19 (Figure 1a) [49]. Given that targeting CD147 has emerged as a promising therapeutic approach for COVID-19, as evidenced by clinical trials [26], we aimed to identify strategies for suppressing CD147. Initially, we investigated the transcriptional/translational regulation of *BSG*. Previous RNP-IP findings indicated a potential interaction between the *BSG* mRNA and HuR [50]. HuR is an RNA-binding protein known to regulate mRNA stability and post-transcriptional processes, and it has been implicated in inflammatory processes by stabilizing the transcription of inflammatory cytokines [51]. Furthermore, elevated cytoplasmic levels of HuR have been observed in lung [52], liver [53], and renal fibrosis [54], suggesting its involvement in promoting fibrosis in various tissues. To investigate this regulatory potential, we generated HuR knock-out cell clones (from the breast tumor cell line MDA-MB-231, which has been shown to have a high level of HuR) using CRISPR/Cas9 technology [55]. As anticipated, the depletion of HuR in MDA-MB-231 cells resulted in a considerable reduction in CD147 levels (Figure 1b). Additionally, we employed the inducible doxycycline TET-off system to overexpress HuR in Hela cells. Consistent with previous findings, HuR overexpression led to an increase in CD147 protein levels (Figure 1c).

To confirm the binding and regulatory role of HuR on *BSG* mRNA in respiratory tissue, we conducted RNP-IP assays in respiratory tissue cell lines, namely WI-38, NL20, and lung epithelial carcinoma H460. By precipitating HuR and the associated RNA, qPCR analysis revealed that *BSG* mRNA exhibited significantly higher enrichment in the HuR group compared to the IgG control in these cell lines. *TGFB1*, a well-characterized HuR binding target, served as a positive control in this assay (Figure 1d–f). These findings provide evidence that HuR can bind to *BSG* mRNA through its 3′-UTR.

Furthermore, to further validate the binding between HuR and *BSG* mRNA, we performed an RNA-IP assay. The *BSG* 3′-UTR was chosen based on the previous RNP-IP database [50]. Random RNA oligos, non-biotinylated *BSG* 3′-UTR oligos, and biotinylated *BSG* 3′-UTR RNA oligos were employed to assess the ability of HuR to bind to *BSG* mRNA (Figure 1g). In bronchial epithelial NL20 and lung fibroblast WI-38 cells, HuR strongly bound to biotinylated *BSG*-3′-UTR oligos compared to random oligos. Additionally, when HuR was overloaded with a competing oligo (non-biotinylated *BSG* 3′-UTR), the binding of HuR to biotinylated *BSG* 3′-UTR was significantly reduced. These results confirm that HuR binds to the 3′-UTR of *BSG* mRNA, potentially facilitating its post-transcriptional regulation. Taken together with the results obtained from HuR knock-out clones and the TET-off HuR overexpression clone, these findings provide conclusive evidence that HuR upregulates CD147 in a post-transcriptional manner, thereby promoting the levels of CD147 protein.

### 3.2. Niclosamide Inhibits HuR Nucleocytoplasmic Translocation

Under homeostasis, HuR is primarily localized in the nucleus, while a small fraction of RNA-bound HuR is transported to the cytoplasm to regulate the translation of target mRNAs [56]. It has been demonstrated that niclosamide inhibits the translocation of NF-κB from the cytoplasm to the nucleus by preventing its phosphorylation [57]. This led us to investigate whether niclosamide could also inhibit the nucleocytoplasmic translocation of HuR. Next, to assess the inhibitory effect of niclosamide on HuR translocation, we performed immunocytochemical (ICC) staining in respiratory cell lines with or without the addition of 1μM niclosamide. In the control sample, we observed weak HuR signals in the cytoplasm of WI-38 and NL-20 cells, which is consistent with previous reports [56] (Figure 2a,a’,c,c’, highlighted with arrows). However, upon niclosamide treatment, there was a notable reduction or absence of HuR signal in the cytoplasm, indicating an inhibition of HuR cytoplasmic translocation (Figure 2b,b’,d,d’). Furthermore, we conducted immunohistochemical (IHC) staining on LL/2 tumor tissues after intraperitoneal injection of 10 mg/kg niclosamide or PBS, administered daily for 5 days. In vivo IHC staining of mouse LL/2 tumors (Figure 2e,e’) indicated a suppression of HuR cytoplasmic translocation by niclosamide as well.

To further verify the inhibitory effect of niclosamide on HuR translocation, we treated WI-38, NL20, and H460 cells with various concentrations of niclosamide ranging from 0 to 1 μM for 16h. Consistent with the ICC findings, the levels of cytoplasmic HuR were significantly reduced with increasing concentrations of niclosamide. We observed a reduction of nearly 30% in NL20 cells (Figure 3a,a’), 50% in H460 cells (Figure 3c,c’), and up to 75% in WI-38 cells (Figure 3e,e’). Furthermore, we examined the time-dependent effect of niclosamide on HuR translocation over a period of 0 to 20 h in 4 h intervals. As anticipated, niclosamide inhibited HuR translocation in a concentration-dependent manner within these cell lines at a concentration of 1 μM. We observed a decrease of approximately 30% in NL20 cells (Figure 3b,b’), 60% in H460 cells (Figure 3d,d’), and 75% in WI-38 cells (Figure 3f,f’). These results clearly demonstrate that niclosamide effectively suppresses the nucleocytoplasmic translocation of HuR.

### 3.3. Niclosamide Significantly Reduces Overall CD147 Protein Levels

Recent research has unveiled the CD147-spike protein as a novel pathway for SARS-CoV-2 to invade host cells, emphasizing the potential of CD147 as a promising target for developing effective drugs against COVID-19 [12]. Furthermore, studies have indicated that CD147, a glycoprotein, exists in two different glycosylation forms: high glycosylated (HG) and low glycosylated (LG) [58]. Notably, glycosylation of CD147 plays a significant role in pathological cardiac hypertension and fibrosis. Overexpression of the highly glycosylated HG form, but not the non-glycosylated CD147 mutant, has been shown to increase cardiac fibrosis, exacerbate cardiac hypertension, and worsen myocardial oxidative stress and ferroptosis [59].

To identify potential therapeutic agents for COVID-19, we investigated the impact of niclosamide on CD147 based on our newly discovered HuR-CD147 regulatory mechanism. The results demonstrated that niclosamide effectively reduced CD147 protein levels in various cell lines, including HEK-293FT, Hela, colon carcinoma cell line RKO, and MDA-MB-231 (Figure 3g), as well as respiratory cell lines such as WI-38, NL20, and H460 (Figure 3a–f). Furthermore, niclosamide exhibited a pronounced dose- and time-dependent reduction in CD147 levels across multiple cell lines (Figure 3a–f). Notably, the CD147 levels were further decreased by niclosamide in MDA-MB-231 HuR knock-out clones (Figure 1b,b’), and the suppression was partially rescued by HuR overexpression (Figure 1c,c’), indicating that niclosamide inhibits CD147 in both HuR-dependent and independent manners. These findings confirm the effective and dose- and time-dependent reduction in CD147 levels caused by niclosamide.

### 3.4. Niclosamide Effectively Inhibits SARS-CoV-2-Induced CD147

As mentioned earlier, RNA-seq analysis revealed that SARS-CoV-2 infection upregulates the expression of *BSG* [49]. In this study, we aimed to (a) validate this increase in *BSG* expression and (b) investigate the potential of niclosamide to suppress CD147 in SARS-CoV-2-infected cells. ACE2-expressing A549 cells (A549-ACE2) were treated with niclosamide at concentrations of 0, 0.25, and 0.5 μM. After 2 h, SARS-CoV-2 was added to the cell culture at an MOI of 2. Our results demonstrated that SARS-CoV-2 infection induced an elevation in the HG form of CD147, while the LG form of CD147 was decreased compared to the control group (Figure 4a–c). Importantly, niclosamide effectively hindered both forms of CD147, particularly the HG form (Figure 4b,c). Notably, the levels of both HG and LG CD147 were reduced compared to the control group, suggesting that niclosamide inhibits CD147 by modulating processes that occur prior to post-translational regulation, rather than targeting specific glycosylation forms. These findings not only confirm that SARS-CoV-2 promotes an increase in HG CD147, but also demonstrate the ability of niclosamide to mitigate this increase effectively.

## 4. Discussion

The role of CD147 as a potential entry point for SARS-CoV-2 has been a subject of ongoing debate, with conflicting data presented by different laboratories, leading to ambiguity in defining CD147’s function [12,60]. However, recent clinical trials targeting CD147 have shown promise in accelerating the recovery of COVID-19 patients [26]. Additionally, various studies have highlighted the potential interaction between CD147 and SARS-CoV-2 in megakaryocytes, platelets, human CD147 knock-in NSG mice models, and iPSC-derived kidney podocyte models [19,20,22,23]. Furthermore, considering CD147’s involvement in fibrosis progression [61,62,63], particularly in its HG form [59], these findings collectively support the notion that CD147 could be a valuable target for mitigating COVID-19 disease and post-COVID-19 conditions. In this study, we discovered HuR promotes *BSG* expression by binding to the 3′-UTR of its mRNA. Furthermore, we observed that niclosamide inhibits the cytoplasmic translocation of HuR and reduces CD147 across various cell lines. Additionally, we found that niclosamide effectively suppresses the upregulation of HG-CD147 induced by SARS-CoV-2.

Interestingly, niclosamide exhibits different effects on HuR translocation at low and high concentrations. At lower concentrations, niclosamide effectively reduces the translocation of HuR from the nucleus to the cytoplasm. However, this effect is reversed in certain cell lines when the concentration of niclosamide exceeds 1 μM. This reversal could be attributed to the induction of endoplasmic reticulum stress by higher concentrations of niclosamide, which triggers the translocation of HuR to the cytoplasm (manuscript in preparation). In this study, we identify niclosamide as an inhibitor of HuR nucleocytoplasmic translocation. The exact mechanism by which niclosamide counteracts HuR translocation remains unknown. Although we did not observe a strong direct binding between HuR and niclosamide, it is reasonable to assume that niclosamide might indirectly affect the phosphorylation or dimerization of HuR, both of which are required for proper translocation [25]. Considering that HuR is also involved in fibrosis and the expression of inflammatory cytokines [57], further investigation into the combined effect of HuR/CD147 inhibition with niclosamide would be advantageous.

Furthermore, it is noteworthy that SARS-CoV-2 specifically induces the HG form of CD147 compared to the control, which has direct implications for the occurrence of post-COVID-19 cardiac conditions. The spike protein of the virus, being heavily glycosylated, requires proper glycosylation for its entry into host cells [64,65]. This association between SARS-CoV-2 infection and the upregulation of HG CD147 may help elucidate the cardiovascular complications observed in the later stages of COVID-19 [59]. Encouragingly, our findings demonstrate that niclosamide can effectively reduce the protein levels of CD147, which has been identified as a promising therapeutic target, and suppress the HG form of CD147. This provides another potential avenue for treating COVID-19 and its associated conditions.

Given the emergence of new SARS-CoV-2 variants with enhanced immune evasion against vaccines [66] and the growing prevalence of long-term lung damage and cardiovascular complications resulting from COVID-19 [67], there is an urgent need for drugs targeting COVID-19 and its related conditions. In this study (Figure 4d), we not only uncover a novel mechanism for regulating CD147 through the RNA-binding protein HuR but also demonstrate that niclosamide effectively inhibits HuR nucleocytoplasmic translocation, reduces CD147 protein levels, and prevents the increase in CD147 upon SARS-CoV-2 infection. These findings serve as a proof of principle for repurposing niclosamide as a functional inhibitor of CD147 and highlight its potential as a therapeutic option for COVID-19.

## 5. Conclusions

In our present study, we have uncovered a significant finding that highlights the binding of HuR to the 3′-UTR of *BSG* mRNA, leading to the upregulation of CD147 protein levels. This discovery identifies CD147 as a target of HuR, shedding light on its regulatory role. Importantly, we have demonstrated that niclosamide effectively reduces CD147 levels through both HuR-dependent and independent mechanisms. These findings firmly establish niclosamide as a promising inhibitor of CD147.

Furthermore, multiple research groups have also recognized the potential therapeutic value of targeting CD147 in the context of COVID-19, and niclosamide has already exhibited high efficacy against SARS-CoV-2 in previous studies. Additionally, niclosamide has shown suppressive effects on fibrosis, further enhancing its potential in combating COVID-19 disease progression. Collectively, our results unveil a novel mechanism by which niclosamide can contribute to the management of COVID-19.

## Figures and Tables

**Figure 1 biomedicines-11-02019-f001:**
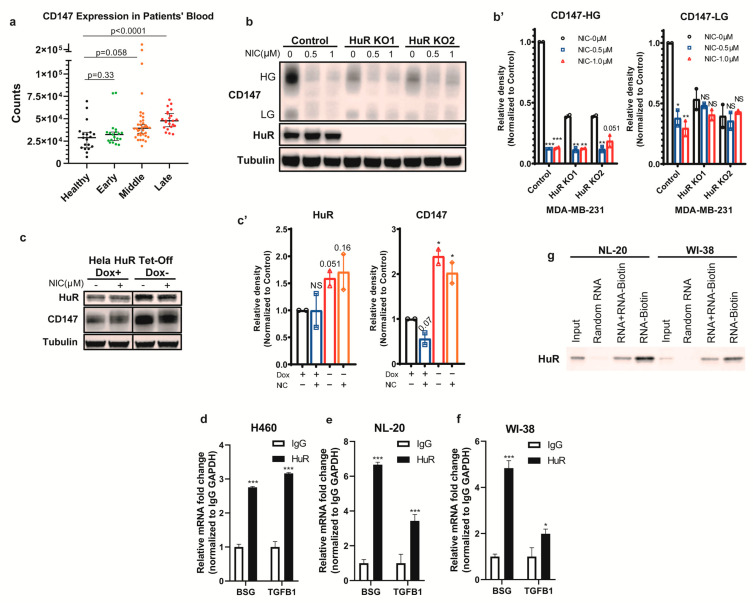
HuR binds *BSG* mRNA and regulates its expression. (**a**) CD147 expression in COVID-19 patients’ blood samples. Here, 6 out of 19 early patients, 6 out of 36 middle patients, and 0 out of 22 late patients are severely ill (the rest are considered as mild/moderate). (**b**) Western blotting analysis of HuR and CD147 protein in HuR knock-out clones of MDA-MB-231 cells with or without niclosamide treatment. Tubulin was used as the loading control. NIC, niclosamide. (**b’**) Quantified relative level of CD147 HG and LG in (**b**). (**c**) Western blotting analysis of HuR and CD147 protein in doxycycline-inducible HuR TET-off system in Hela cells. Dox, doxycycline. (**c’**) Quantified relative level of CD147 and HuR in (**c**). (**d**–**f**) RNP-IP analysis of relative enrichment of *BSG* transcripts in HuR-immunoprecipitation in (**d**) H460, (**e**) NL20, and (**f**) WI-38 cells. (**g**) Western blotting of HuR protein in the pull-down complex by *BSG* RNA probes in NL20 and WI-38 cells. *** *p* < 0.001; ** 0.001 < *p* < 0.01; * 0.01 < *p* < 0.05. NS, not significant. Two-tailed *T*-test was applied for statistics.

**Figure 2 biomedicines-11-02019-f002:**
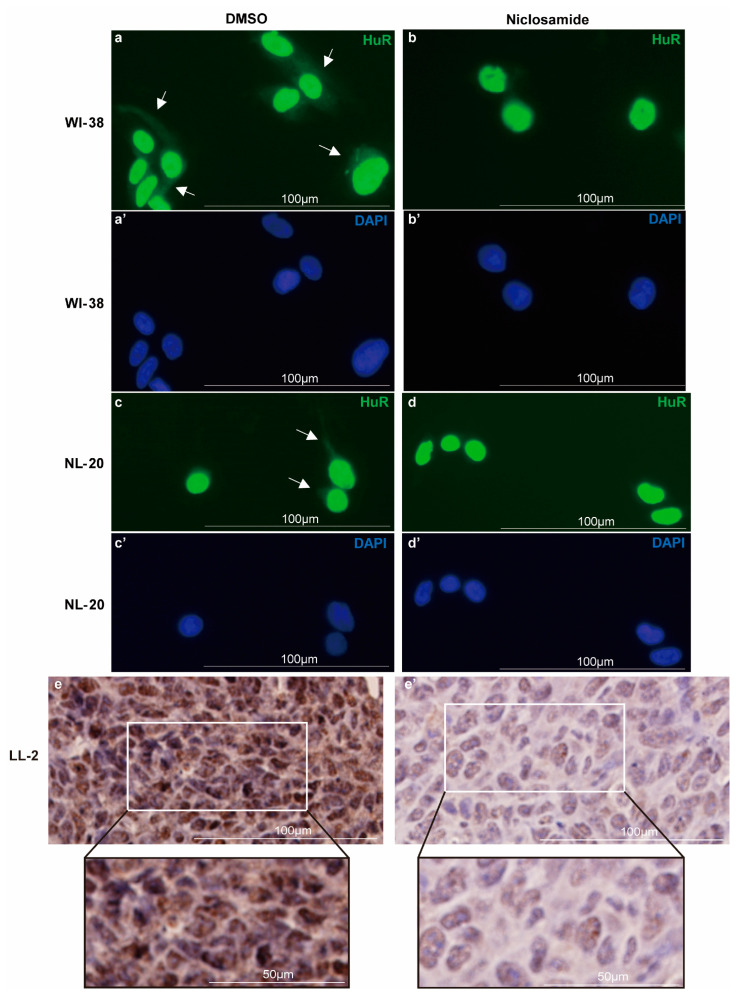
Immunocytochemistry and immunohistochemistry staining for HuR and the effect of niclosamide on HuR translocation. (**a**–**d’**) Immunofluorescence staining for the detection of HuR and the effect of niclosamide on HuR cytoplasmic translocation. WI-38 (**a**–**b’**) and NL20 (**c**–**d’**) cells were exposed to niclosamide or DMSO for 48 h. HuR is presented in green. Nuclei were counterstained with DAPI (blue). (**e**,**e’**) Immunohistochemistry staining for HuR protein in the LL/2 tumor tissue with (**e**) PBS or (**e’**) niclosamide treatment for 5 days. Arrows point to the cytoplasmic HuR.

**Figure 3 biomedicines-11-02019-f003:**
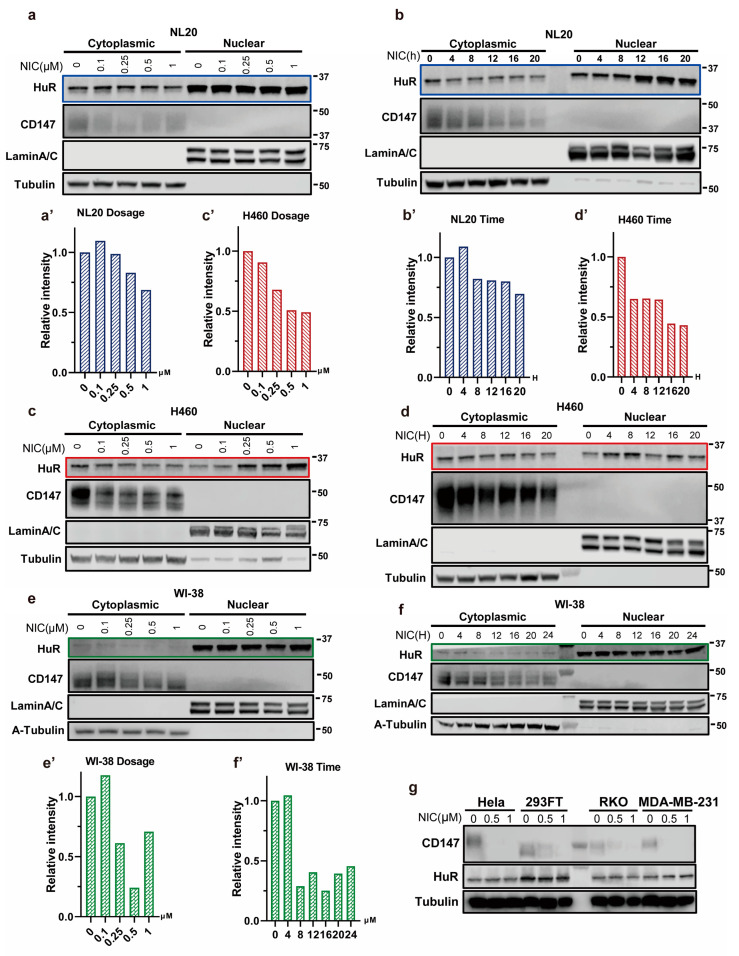
Niclosamide inhibits HuR nucleocytoplasmic translocation. (**a**–**f**) Western blot of HuR and CD147 in nuclear and cytoplasmic fractions with niclosamide treatment at different concentrations and different times in NL20 (**a**,**b**), H460 (**c**,**d**), and WI-38 (**e**,**f**). Quantified relative levels of cytoplasmic HuR in NL20 (as in blue box), H460 (as in red box), and WI-38 (as in green box) are shown in (**a’**–**f’**), respectively. (**g**) Western blot of HuR and CD147 protein in Hela, 293FT, RKO, and MDA-MB-231 cells treated with niclosamide at different concentrations. NIC, niclosamide.

**Figure 4 biomedicines-11-02019-f004:**
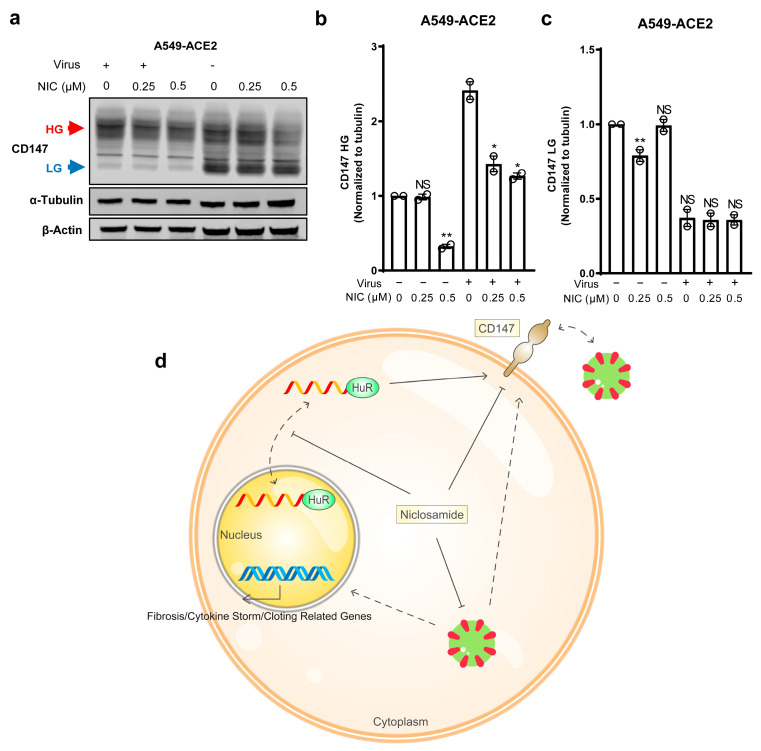
Niclosamide effectively inhibits SARS-CoV-2-induced CD147. (**a**) Western blotting analysis of CD147 protein in A549-ACE2 cells with or without SARS-CoV-2 infection and niclosamide treatment. Spike protein was used as the virus marker and tubulin was the loading control. (**b**,**c**) Quantification of relative band intensity of HG CD147 and LG CD147. (**d**) The proposed working model of niclosamide restricting SARS-CoV-2 in vitro. Schematic diagram of the mechanism: niclosamide disrupts the nucleocytoplasmic shuttling of HuR, thereby inhibiting the post-transcription of CD147 mRNA and abolishing the glycosylation and maturation of CD147. In addition, niclosamide effectively restrains SARS-CoV-2-induced CD147. Therefore, niclosamide can be a promising anti-COVID-19 drug by blocking HuR nucleocytoplasmic translocation and SARS-CoV-2 entry route CD147. ** 0.001 < *p* < 0.01; * 0.01 < *p* < 0.05. NS, not significant. Two-tailed T-test was applied for statistics.

## Data Availability

Not applicable.

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
