# Peer review of "Repurposing Niclosamide as a Novel Anti-SARS-CoV-2 Drug by Restricting Entry Protein CD147"

_biomedicines, 2023, doi:10.3390/biomedicines11072019_

Round 1

Reviewer 1 Report

In this paper, the authors explored the modulation of CD147 by HuR protein (a RNA binding protein located in the nucleus of the cells). The demonstrated that HuR binds to CD147 mRNA, and the depletion of HuR results in a decrease in CD147 protein expression. Starting from the consideration that niclosamide can inhibit the translocation of NF-κB from the cytoplasm to the nucleus by preventing its phosphorylation, they demonstrated that niclosamide can also inhibit the translocation of HuR from the nucleus to the cytoplasm and so it can be used to inhibit CD147 and consequently as a potential drug for COVID-19 (since CD147 has also been implicated in collateral effect produced by the SARS-CoV-2 spike protein).

The conclusions about the correlation between HuR and CD147 concentration and about the efficacy of niclosamide to inhibit HuR translocation are supported by the presented data.

I suggest two modifications to improve the quality of the manuscript:

-      -  the Immunocytochemistry results reported in Figure 2 are not clearly presented in the text, so it is difficult to follow what the pictures mean (please explain how the technique works and the information that can be derived by the different pictures)

-      -  The discussion paragraph reports information from literature and the correlation between literature data and the data reported in this paper sometimes is not clear. So please re-write this paragraph focusing on the results reported in the paper.

Reviewer 2 Report

The manuscript titled “Repurposing niclosamide as a novel anti-SARS-CoV-2 drug by restricting entry protein CD147” by Yang, Z.; is an original work where the authors assess the niclosamide binding and subsequent reduction of CD147 expression. Many alternatives techniques like western-blotting, immunofluorescence staining, protein/nucleic acid coimmunoprecipitation, and cell survival assays were used in order to better undertand the underlying mechanisms of this binding process. The study is interesting and it is well-designed.

However, it exists some points that need to be addressed (please, see them below detailed point-by-point). The most relevant outcomes found by the authors can contribute to design more efficient therapies to SARS-CoV-2 disease. Furthermore, the methodological approach used by the authors could be expandable for other diseases where protein membrane receptors are involved. For this reason, I will recommend the present scientific manuscript for further publication in Biomedicines once all the below described suggestions will be properly fixed.

Here, there exists some points that must be covered in order to improve the scientific quality of the manuscript paper:

1) ABSTRACT. “Here, we demonstrate (…) 3´-UTR” (lines 20-22). Please, the authors should define the term “UTR” by adding “untranslated trailer region”. Then, the abbreviation should appear between brackets.

2) KEYWORDS (OPTIONAL). It may be desirable to add the most relevant techniques used in this study like the immunohistochemistry staining.

3) INTRODUCTION. “Currently, only three drugs have gained approval from the U.S. Food and Drug Administration (…) for patients with renal disease” (lines 40-46). Here, the authors point out the existing available drugs from the FDA and their potential side effects. It is also neccesary to indicate the FDA adverse reporting system (FAERS) which is currently under study to better evaluate potential adverse reactions suffered by the patients [1].

[1] Laurini, G.S.; Montanaro, N.; Motola, D. Safety Profile of Monupiravir in the Treatment of COVID-19: A Descriptive Study Based on FAERS Data. J. Clin. Med. 2022, 12, 34. https://doi.org/10.3390/jcm12010034. 

4) MATERIALS & METHODS. “(…) virus titer is 1.0 x 107 plaque (…)” (line 189). Please, the authors should add the superscript to the information provided in this statement.

5) A549-ACE2 cells (…) 0, 0.25, 0.5 µM. Please, the authors should homogenize the significant figures. This point should be covered in the rest of the main manuscript body text.

6) RESULTS. Figure 1, panels (b´), (c´), (d), (e) and (f) (line 241). The authors should devote statistical analysis (e.g. Student’s t-test, or analysis of variance (ANOVA)) in order to know if the observed differences are statistically significant. Same comment for the Figure 4, panels (b) and (c) (line 331).

7) Figure 2, panels (e) and (e´) (line 266). Please, the authors should add the relative scale bars to the zoomed inset images. Furthermore, the authors should specify the meaning of the white arroys appeared in the upper image in the respective figure caption.

8) DISCUSION. “Additionally, various studies have highlighted the potential interaction between CD147 and SARS-CoV-2 (…) models” (lines 346-349). Here, the authors show previously reported works where the interactions between CD147 and SARS-CoV-2 are addressed. Nevertheless, it may be desirable to furnish information about complementary techniques to gather this knowledge to pave the way in future perspectives like Förster resonance energy transfer (FRET) [2], atomic force microscopy based force spectroscopy (AFM-FS) [3] or fluorescence cross-correlation microscopy (FCCS) [4].

[2] Rainey, K.H.; Patterson, G.H. Photoswitching FRET to monitor protein-protein interactions. Proc. Natl. Acad. Sci. U. S. A. 2019, 116, 864-873. https://doi.org/10.1073/pnas.1805333116.

[3] Lostao, A.; Lim, K.; Pallares, M.C.; Ptak, A.; Marcuello, C. Recent advances in sensing the inter-biomolecular interactions at the nanoscale – A comprehensive review of AFM-based force spectroscopy. Int. J. Biol. Macromol. 2023, 238, 124089. https://doi.org/10.1016/j.ijbiomac.2023.124089.

[4] Fukimoto, A.; Lyu, Y.; Kinjo, M.; Kitamura, A. Interaction between Spike Protein of SARS-CoV-2 and Human Virus Receptor ACE2 Using Two-Color Fluorescence Cross-Correlation Spectroscopy. Appl. Sci. 2021, 11, 10697. https://doi.org/10.3390/app112210697.

9) CONCLUSIONS. This section is clear and concise. The authors highlights the most relevant outcomes found in this work. No actions are requested to the authors.

10) REFERENCES. The references are not in the proper format style of Biomedicines. The publication name should not appear in Italics, the journal name should appear in abbreviated form and remarked in Italics, the publication year should be highlighted in bold, etc. Please, the authors should fix this point.

English is fine. Nevertheless, the authors should recheck the manuscript in order to polish some final aspects susceptible to be improved.

Reviewer 3 Report

Manuscript Title: Repurposing niclosamide as a novel anti-SARS-Cov-2 drug by restricting entry protein CD147

Reviewer Recommendation: Major  revision

Comments to the Author

1. Authors requested to provide detailed information on the range of concentration of the compounds and how they are fixed concentrations for each experiment?

2. Authors have mentioned the correlation of CD147 in four different stages of COVID-19 patients, but no information on the patients were infected with mild or moderate or severe, if the author provides this information it could be more informative.

3. Authors requested to provide detailed abbreviation

4. In the methods section it was mentioned that a number of cancer cell lines were used in this study, but in the results section CD147 expression was studied only on MDA-MB-123, what about the other cell lines?

5. What basis authors have fixed the concentration of niclosamide ranging from 0 to 1 μM for 16h, for immunofluorescence assay?

6. How have authors incorporated the results of expression of CD147 on cancer cell line with SARS-CoV-2 infection?

7. Authors requested to provide information on in vivo studies more in detail.

8. Authors requested to increase the figure quality. 

Some minor edits in language is required
